# Antiseptic, Hemostatic, and Wound Activity of Poly(vinylpyrrolidone)-Iodine Gel with Trimethyl Chitosan

**DOI:** 10.3390/ijms25042106

**Published:** 2024-02-09

**Authors:** Andrew Padalhin, Hyun Seok Ryu, Seung Hyeon Yoo, Celine Abueva, Hwee Hyon Seo, So Young Park, Phil-Sang Chung, Seung Hoon Woo

**Affiliations:** 1Beckman Laser Institute Korea, Dankook University College of Medicine, Cheonan 31116, Republic of Korea; drdrew200708@dankook.ac.kr (A.P.); ryuhs2023@dankook.ac.kr (H.S.R.); cgabueva@dankook.ac.kr (C.A.); ryuhs@dankook.ac.kr (S.Y.P.); pschung@dankook.ac.kr (P.-S.C.); 2School of Medical Laser, Dankook University, Cheonan 31116, Republic of Korea; yoosh6653@dankook.ac.kr (S.H.Y.); shh8881@dankook.ac.kr (H.H.S.); 3Medical Laser Research Center, Dankook University, Cheonan 31116, Republic of Korea; 4Department of Otorhinolaryngology-Head and Neck Surgery, Dankook University College of Medicine, Cheonan 31116, Republic of Korea

**Keywords:** antiseptic, hemostatic, wound healing, trimethyl chitosan

## Abstract

Wound management practices have made significant advancements, yet the search for improved antiseptics persists. In our pursuit of solutions that not only prevent infections but also address broader aspects of wound care, we investigated the impact of integrating trimethyl chitosan (TMC) into a widely used poly(vinylpyrrolidone)-iodine gel (PVP-I gel). Our study assessed the antimicrobial efficacy of the PVP gel with TMC against *Escherichia coli*, *Staphylococcus aureus*, multidrug-resistant *S. aureus* MRSA, and *Candida albicans*. Additionally, we compared hemostatic effects using a liver puncture bleeding model and evaluated wound healing through histological sections from full-thickness dermal wounds in rats. The results indicate that incorporating TMC into the commercially available PVP-I gel did not compromise its antimicrobial activity. The incorporation of TMC into the PVP-I gel markedly improves its hemostatic activity. The regular application of the PVP-I gel with TMC resulted in an increased blood vessel count in the wound bed and facilitated the development of thicker fibrous tissue with a regenerated epidermal layer. These findings suggest that TMC contributes not only to antimicrobial activity but also to the intricate processes of tissue regeneration. In conclusion, incorporating TMC proves beneficial, making it a valuable additive to commercially available antiseptic agents.

## 1. Introduction

Over a lifetime, the human body inevitably experiences numerous instances of trauma and injuries, challenging its physical and physiological well-being. These injuries manifest as acute or chronic wounds, breaching the protective skin barrier and exposing underlying tissues and cells, rendering them susceptible to potentially harmful microbial infections [1,2,3,4,5,6]. In light of this vulnerability, a cautious and diligent approach to wound care becomes imperative.

When addressing wounds, special attention must be given to the delicate balance required in the cleansing process [7,8]. The meticulous removal of dirt and debris from affected tissues is necessary to prevent inadvertent harm. Resorting to aggressive chemicals or forceful physical manipulations during wound cleansing should be avoided at all costs [9,10]. While well intentioned, such interventions carry the risk of inflicting additional trauma to already compromised organs and tissues, thereby impeding the natural healing process.

Minimizing disruption to wounds is not solely about alleviating immediate pain and discomfort; it plays a pivotal role in mitigating the potential for further tissue damage and the onset of secondary infections [11,12]. Disturbing the healing environment can set the stage for complications, leading to a protracted recovery period. Consequently, adopting practices that promote minimal interference becomes paramount in safeguarding the overall well-being of the patient [13]. Failure to implement appropriate wound management practices can have far-reaching consequences [7,9]. What may initially present as a simple wound has the potential to deteriorate over time, adversely impacting the patient’s quality of life [7,13]. Furthermore, neglected wounds can contribute to significant morbidity, underscoring the critical importance of timely and effective interventions in ensuring optimal outcomes [4,5].

Apart from the continuous development of biocompatible hydrogel dressings [14,15,16], the requisites for effective wound care have given rise to the widespread utilization of antiseptic agents in the field of wound management, particularly in maintaining aseptic conditions within surgical settings. One such widely accepted antiseptic agent is polyvinylpyrrolidone iodine (PVP-I), commonly recognized as povidone-iodine [17,18,19,20,21]. PVP-I exhibits its antimicrobial prowess by releasing free iodine, which interacts with enzymes and proteins, rendering them inactive [5,17,18,19,21]. This attribute has made PVP-I a versatile solution, as it is soluble in water and adaptable for various applications, including but not limited to functioning as a first aid antiseptic, pre-operative germicidal scrub, surgical disinfectant, and as a tincture or ointment for treating infected wounds [17,18,19,20,21,22,23,24,25]. Despite its extensive use in clinical and emergency settings, the existing literature on the impact of PVP-I on cells and wound healing remains inconclusive. Some reports enthusiastically highlight the benefits of PVP-I, underscoring its positive impact on wound conditions [5,18,19,24], while others suggest limited or negligible effects in terms of antiseptic activity [26,27,28] or wound tissue development [29,30,31,32,33,34]. This divergence in findings necessitates the continued exploration of different formulations and modifications that can enhance the efficacy of PVP-I in wound management.

Recent studies have taken a step further by exploring the combination of PVP-I with various biomaterials and antiseptics, aiming to potentially amplify its effectiveness as an antiseptic agent [35,36,37]. This evolving line of research holds promise in uncovering novel approaches to optimize the therapeutic potential of PVP-I, shedding light on its nuanced interactions with cells and tissues during the wound healing process. As the scientific community delves deeper into these investigations, a more comprehensive understanding of PVP-I’s multifaceted role in wound management is expected to emerge, providing valuable insights for refining its applications and maximizing its benefits in clinical practice. Among the broadly investigated biomaterials for wound healing is chitosan, a natural carbohydrate polymer derived from chitin, the main component of crustacean exoskeletons [38]. Chitosan is a highly deacetylated form of chitin and is primarily soluble in acidic aqueous media [38,39,40]. This polymer has found its way into several industries such as pharmaceutics, food and beverage, water treatment, agriculture, and cosmetics because of its biodegradability and biocompatibility [38]. By adding methyl groups to the existing amino groups of chitosan and converting its primary amino groups into quaternary ammonium groups, a positively charged water-soluble form of chitosan can be produced—trimethyl chitosan (TMC). This highly modified chitosan derivative can interact with negatively charged drug molecules or interact with DNA or RNA molecules, making it suitable for improving stability and facilitating drug and gene delivery through improved bioavailability. TMC is also known to be mucoadhesive and thus suitable for topical applications [41,42,43,44,45]. Like PVP-I, relevant research on trimethyl chitosan has also disclosed its antimicrobial potential [41,45,46]. 

Although the general use of povidone-iodine as an antiseptic is widely accepted, there is still room for improvement in its application. This prospective study aims to provide information on the improved performance of a commercially available polyvinylpyrrolidone iodine (PVP-I) gel mixed with trimethyl chitosan (TMC) concerning several aspects of wound management. The antimicrobial activity, hemostasis, wound size reduction, and wound tissue development of the TMC-modified PVP-I gel were compared with other existing formulations of PVP-I.

## 2. Results

Four types of commercially available iodophor antiseptics were tested in this study. The test samples and corresponding name code for the study are as follows: 3% PVP-I solution (PI3-S) (Firson Co., Ltd., Cheonan, Republic of Korea); 3% PVP-I gel (PI3-G) (Firson Co., Ltd., Cheonan, Republic of Korea); Guard W-gel, 3% PVP-I gel with TMC (PI3-G+TMC) (Firson Co., Ltd., Cheonan, Republic of Korea); and Repigel^®^, 3% PVP-I liposomal hydrogel (PI3-LH) (Mundipharma Research GmbH & Co., Cambridge, UK). 

### 2.1. Antiseptic Activity

The various povidone formulations underwent testing against different bacterial and fungal species commonly found as contaminants in dermal wounds [47,48,49]. Microbial viability was assessed at two observation points, specifically 1 h and 24 h after inoculation. Figure 1 displays the viability of *E. Coli*, *S. aureus*, MRSA, and *C. albicans*. The results suggest that all test samples effectively reduced bacterial and fungal viability within 1 h (Figure 1B). As anticipated, PI3-S exhibited the lowest viability among all samples, with an average microbial viability reduction of 80–75%. Both PI3 g and PI3-G+TMC demonstrated a similar performance, with a 60% and 58% average microbial viability reduction, respectively. PI3-LH showed modest microbial viability reduction across all samples, with an average of 30–35%. A comparison of microbial viability 24 h after inoculation revealed differential antiseptic activity among the tested samples (Figure 1C). There was a further reduction in bacterial and fungal viability in cultures treated with PI3-S, PI3-G, and PI3-G+TMC, indicating bactericidal/fungicidal activity. In contrast, PI3-LH generally maintained microbial viability relatively close to its results 1 h after inoculation, typical of a bacteriostatic/fungistatic agent [50,51] relative to the other samples.

### 2.2. In Vivo Hemostasis

The current study also compares the hemostatic activity of PVP-I gel with TMC to other commercially available PVP-I gel formulations. The hemostatic capacity of each test sample was evaluated using an established in vivo liver punch bleeding model. Based on timed video recordings, the application of both saline solution and PI3 g did not significantly differ from the application of normal saline solution, resulting in a relatively similar average bleeding time of 278.7 and 262.9 s, respectively (Figure 2—inset video still); However, PI3-S yielded a considerably lower average bleed time (176.4 s) compared to both saline and PI3 g applications. Additionally, PI3-LH had an average bleeding time of 58.1 s, while PI3-G+TMC had 63.3 s, both drastically reduced compared to the application of saline, PI3-S, and PI3-G. Representative video files can be viewed in the provided Appendix A. According to the results of the in vivo hemostatic test, the addition of TMC in the PVP-I gel markedly improved its capability to stop bleeding, making it comparable to PI3-LH. Furthermore, these results indicate that the hemostatic activity of these types of antiseptics is based on the added component rather than the viscosity of the resulting material. 

### 2.3. In Vivo Wound Healing

The study also observed the impact of regularly applying various samples onto open full-thickness dermal excision wounds in rat animal models. Macro images of the wound bed were captured every other day for one week, serving as a reference for assessing size reduction over time. A general observation from the treated wounds indicated that the application of PI3-S and PI3-LH resulted in the immediate dehydration of the wound area, while the application of PI3 g and PI3-G+TMC remained slightly moist 24 h after each application session (Figure 3A). Measurements of the wound size at different time points did not reveal any significant difference among the treatments, suggesting that the contraction rate and wound closure were consistently similar across the treatment groups (Figure 3B). Although the comparison of wound size did not show any significant difference among the treatment groups, histological examinations of the extracted tissues revealed that routine exposure to the different antiseptic agents led to distinct tissue regeneration. 

#### 2.3.1. Tissue Blood Vessel Count

High-magnification images were captured from the stained tissue sections and seamlessly stitched to create composite images for blood vessel counting in each tissue sample. Using Fiji software (version 2.9.0), blood vessels were meticulously marked and counted along the wound margin and within the fibrous tissue area of each tissue section. In Figure 4A, representative images showcase the general distribution of blood vessels within the tissue sections from various treatment groups. Among the samples analyzed, the group treated with saline exhibited the lowest number of capillaries, with an average of 26 per tissue section. Following this, groups treated with either PI3-S or PI3-LH showed an average blood vessel count of 45 and 42, respectively. The PVP-I gel displayed an average of 53 blood vessels per tissue section, while PI3-G+TMC demonstrated a higher count with an average of 78 per tissue section (Figure 4B). These findings suggest that the application of PI3-G+TMC better promoted the formation of microvascular networks compared to all other samples.

#### 2.3.2. Fibrous Tissue Thickness

In addition to the blood vessel count, tissue samples from various treatment groups were also compared in terms of fibrous tissue development. The thickness of the fibrous tissue formed within the skin defect area was measured, and the average thickness was calculated for each group. Figure 5A displays representative images of the tissue sections and the measurements taken to determine the average fibrous tissue formation from each group. Data collected from the fibrous tissue measurement indicate that PI3-G+TMC exhibited the thickest fibrous tissue formation among all treatment groups, averaging around 589.8 µm. The application of PI3-S, PI3-G, and PI3-LH resulted in similar fibrous tissue thicknesses, measuring 509.5 µm, 474.5 µm, and 471.3 µm, respectively. Saline treatment led to the development of relatively thinner fibrous tissue, averaging 420.1 µm compared to all other samples (Figure 5B).

#### 2.3.3. Re-Epithelialization

Another histological parameter compared among the treatment groups was the thickness of the regenerated epidermis. In Figure 6A, representative images of the regenerated epidermis from different treatment groups are displayed. The regenerated epidermis was defined as the epidermal tissue directly above the fibrous tissue, which does not extend beyond the edges of the wound bed. Unsurprisingly, routine treatment with normal saline solution had little effect on the development of thick regenerated epidermis; however, the application of the antiseptic agents generally resulted in the formation of thicker epidermal tissue within the wound area. The cross-sectional measurements in Figure 6B show that, on average, the routine application of PI3-LH (93.24 µm) and PI3-G+TMC (101.58 µm) led to a thicker regenerated epidermis compared to both PI3-S (75.25 µm) and PI3 g (79.76 µm).

## 3. Discussion

The primary objective of the current study was to determine the impact of integrating trimethyl chitosan (TMC) into commercially available polyvinylpyrrolidone-iodine (PVP-I) gel in the context of its utility for wound management applications. PVP-I stands out as a well-established antiseptic widely employed in clinical settings. Composed of polyvinylpyrrolidone and iodide, forming a complex through hydrogen bonds between two pyrroles, PVP-I serves as a soluble carrier reservoir of active free iodine [5,25]. Given its size, free iodine easily infiltrates microorganisms, oxidizing cytoplasmic proteins crucial for their survival. This antimicrobial mechanism imparts relatively low microbial resistance but also lends a hand in limiting PVP-I’s antimicrobial activity against micro-organisms that have high mycolic acid content which prevents free iodine absorption [52]. Under aqueous conditions, there is a continuous release of small amounts of free iodine which remains in dynamic equilibrium within the complex [25]. The typical formulation in commercially available PVP-I is approximately 10% in water, with other lower concentrations ranging down to 2.5%. Interestingly, the bactericidal activity of PVP-I does not directly correlate with concentration but rather follows a bell curve with increasing dilution [53]. This is because PVP-I can release more free active iodine upon dilution, even though it theoretically contains the same amount of iodine at higher concentrations. Early studies have established that the effective bactericidal activity of povidone begins at a 1:10 dilution (10%), which is markedly maintained up to a dilution of 1:100 (0.1%). Beyond this point, a sharp decrease in antibacterial activity is observed [22,53]. In this study, a consistent concentration of 3% was employed across all test groups, aligning with the established antibacterial dilution range of PVP-I. The results from the in vitro antimicrobial tests targeting common wound infections such as *E. coli*, *S. aureus*, MRSA, and *C. albicans* appear unremarkable, with the exception of readings from PI3-LH. Despite TMC’s inherent antibacterial effect [38,43,44,45], its addition to the gel-type PVP-I did not significantly enhance its antimicrobial properties. This indicates that the integration of TMC into PVP-I gel did not impact its bactericidal and fungicidal capabilities, even when subjected to a high microbial load. This outcome is comparable, if not less pronounced, than other types of antiseptics, such as chlorhexidine and alcohol, which have also been used in conjunction with PVP-I to provide dual modes of antimicrobial activity for antiseptic applications. Previous studies have suggested that TMC exerts its antimicrobial activity through its polycationic character, enabling it to interact extensively and disrupt microbial cytoplasmic membranes and walls [54,55,56]. This mode of action is akin to chlorhexidine, rendering both effective against Gram-positive bacteria, Gram-negative bacteria, fungi, and viruses [56,57,58]. On the other hand, the antimicrobial activity of alcohol is based on protein denaturation, proving more effective when diluted with water [59,60]. Unlike chlorhexidine, which can be deactivated upon interaction with anionic compounds and surfactants [61,62], TMC constantly holds a positive charge, imparting relatively prolonged antiseptic activity [54,55,63]. Additionally, chitosan and its derivatives are known to be less toxic [38,64] to mammalian cells compared to both alcohol and chlorhexidine, allowing for their utilization at higher concentrations, which can further enhance their antimicrobial activity in a dose-dependent manner.

The process of wound regeneration involves several overlapping stages, depending on the type and severity of the wound. Initially, wound stabilization is achieved by forming stable blood clots, followed by the inflammatory phase. During this phase, immune cells, predominantly macrophages and neutrophils, remove debris and neutralize foreign materials in the wound bed [1,65,66]. These cells also release chemotactic agents that signal other cells to migrate into the wound area. Within a few days, the wound progresses to the proliferative phase, during which new tissue develops. Fibroblast cells play a crucial role in creating a framework for functional vasculature and epithelial growth. Compared to the earlier stages, the remodeling phase is the lengthiest [1,4,67]. In this phase, the regenerated fibrous tissue undergoes gradual modification and replacement to resemble the original tissue. This is accomplished to restore strength and functionality close to pre-injury conditions. In relation to the aforementioned stages of wound healing, two crucial aspects of wound management were tested and compared across the different formulations of PVP-I—hemostasis and wound tissue development.

Hemostasis, a critical aspect of wound management, demands immediate attention upon injury [2,3,4,6]. Preventing severe blood loss is an immediate concern in deep cuts and lacerations since it could not only pose a substantial obstacle in assessing and stabilizing the patient, but it could ultimately cause death [68,69]. Thus, previous studies have sought to address the urgency of hemostasis and wound bed sterilization by incorporating antimicrobials into hemostatic agents [70,71]. Some accounts have highlighted the potential of PVP-I for the management of bleeding [72,73]. While PVP-I may contribute to controlling bleeding to some extent, it was not originally designed with blood clotting in mind, as evident from our results. Although PI3-S yielded a significantly lower bleeding time compared to PI3 g and saline, it was considerably less consistent in achieving a stable clot compared to both PI3-LH and PI3-G+TMC. Other studies have reported that the application of PVP-I, particularly Betadine, resulted in reduced microcirculation in wound beds [74,75]. On the other hand, solid or viscous materials can act as barriers to blood flow, suggesting that highly viscous agents like PI3-LH and PI3-G+TMC are more effective in promoting hemostasis compared to PI3-S or PI3-G. This is primarily due to the noticeable viscous nature of both samples, which is particularly important in creating a physical barrier for the formation of a stable clot. Recent studies on hemostatic materials have demonstrated that chitosan [48] and quaternized-chitosan-based derivatives can promote blood coagulation by absorbing and accumulating blood cells [43], leading to the activation of the coagulation cascade and reduced coagulation time [42]. The results of the current investigation indicate that both PI3-LH and PI3-G+TMC are suitable materials for simultaneous disinfection and bleeding management.

The effect of the different PVP-I formulations on wound healing was also examined using a relevant animal model of full skin defect. This observation was conducted relative to the tissue formed after the routine application of the respective samples. The results of the histomorphometric analyses revealed distinct tissue development within wounds treated with different PVP-I formulations. Notably, the routine application of PI3-G+TMC during the proliferative phase of dermal wound healing resulted in higher levels of vascularization, fibrous tissue thickness, and regenerated epidermis thickness. These notable differences suggest a more advanced stage of tissue development with the routine application of PI3-G+TMC compared to other treatment groups. Adequate vascularization is a critical feature of the proliferative phase, as it is responsible for maintaining nutrient exchange at the cellular level [2,65,76]. By promoting the development of an extensive network of blood vessels, waste products can be easily eliminated, minimizing inflammation and the formation of excessively thick fibrous tissue. Vascularization also plays a crucial role in sustaining proper signaling factors to regulate cellular behavior and eventual tissue remodeling [2,42,52]. The fibrous tissue formed during the proliferative phase serves as the main scaffolding for dermal cells [1,65,66,67]. Establishing a fibrous tissue framework that spans not only the entire gap of the wound but also the full thickness of the skin tissue, providing an adequate platform for more dermal cells to occupy. This minimizes the need for extensive proliferation to fill in remaining gaps and potentially promotes the progression of the remodeling phase. Lastly, re-epithelialization is one of the key early goals for dermal wound healing. As determined by the histomorphometric analyses, wounds treated with PI3-G+TMC resulted in the formation of a thicker fibrous layer and thicker regenerated epidermis. This is important since the regenerated epidermis directly covers the established scar tissue. It serves as a living protective barrier that ensures an underlying microenvironment suitable for remodeling toward functional skin tissue. 

A factor that may have contributed to the observed distinct tissue formation in the excisional skin defects is the presence of moisture. Previous studies have already demonstrated that the maintenance of a moist environment is a key factor influencing re-epithelialization [77,78]. As observed, PI3-G+TMC retained moisture compared to other samples, potentially providing a more suitable environment for cell proliferation and tissue development. Additionally, since all treatment groups contained the same amount of PVP-I, the difference in tissue formation can be solely attributed to the addition of TMC. Recent studies have showcased the versatility of TMC as both a drug carrier [79,80] and a scaffold component [81,82]. Only a few studies have been focused on investigating the wound healing effects of TMC. Notably, it has been utilized as a carrier for DNA plasmids that encode vascularization growth factors to promote angiogenesis in full-thickness skin defects [83,84,85]. Due to its inherent biocompatibility, chitosan, and its derivatives like TMC, have been proven to advance the growth of granulation tissue, a crucial element for wound healing and vascularization [86,87]. 

It should be noted that the current study exclusively addresses tissue regeneration for the skin. The relative performance of TMC-containing PVP-I gel on other types of tissues, such as bone, muscle, and complex tissues, requires further testing. Nevertheless, the observed outcomes of the current study highlight the advantage and versatility of incorporating trimethyl chitosan into the existing antiseptic formulation of PVP-I gel. The results of this study indicate that the addition of TMC to a commercial formulation of PVP-I can not only provide an additional antibacterial mode of action but also include a hemostatic effect and improved wound healing effects. This modified approach could be a crucial strategy for advancing wound management in this era of rising microbial resistance.

## 4. Materials and Methods

### 4.1. In Vitro Antimicrobial Activity

To assess the efficacy of the different solutions in preventing potential infections, the viability of 3 bacterial species, *Escherichia coli* (KCCM 11234), *Staphylococcus aureus* (KCCM 40881), and multidrug-resistant *S. aureus* (MRSA) (KCCM 12256), as well as one fungal species, *Candida albicans* (KCCM 11282), was tested after exposure to each antiseptic agent. All microbial species tested were obtained from the Korean Culture Center of Microorganisms, Seoul, Republic of Korea. Viability tests were conducted based on a previously established protocol [41,88,89] with slight modification. Briefly, inoculations were performed in a 96-well culture plate by mixing 100 µL of LB Broth (Sigma-Aldrich, St. Louis, MO, USA), 40 µL of a bacterial/fungal solution, and 40 µL of the test sample per well (*n* = 6) which was then placed in a 37 °C incubator. Blank wells containing 140 µL of LB broth and 40 µL of the test sample were also prepared as negative controls. The survival of bacteria/fungi after 1 h and 24 h of incubation was determined by adding 20 µL of thiazolyl blue tetrazolium bromide (MTT) (Sigma-Aldrich, USA) solution at a concentration of 5.0 mg/mL. Absorbance values of the plated bacterial /fungus cultures were measured at 600 nm using a TECAN InfiniteM200 Pro (Tecan Group Ltd., Zürich, Switzerland) plate reader. All absorbance values were normalized by subtracting the readings from the respective blank wells of each test sample. The absorbance values of plated bacterial cultures immediately incubated with MTT solution were used as the reference baseline viability. Percent viability was calculated by dividing the normalized absorption data of each sample by the averaged baseline absorbance values. 

### 4.2. In Vivo Hemostasis

All animal experiments were conducted in accordance with the guidelines set by the Institutional Animal Care and Use Committee at Dankook University (DKU-23-006). The method for hemostatic testing performed in this study was adapted with adjustments from prior research [90,91] on bleeding models. For the hemostasis test, 25 Sprague Dawley male rats weighing approximately 200–250 g (6 weeks old, Orientbio Co., Seongnam, Republic of Korea) were randomly distributed among 5 treatment groups (*n* = 5), with one group treated with saline as a positive control reference. Three rats were housed per cage and placed in a climate-controlled animal room with a 12-hour light–dark cycle. Food and water were provided constantly. The hemostasis experiment was conducted after a week of acclimatization. Rats were anesthetized via the inhalation of 2.0–3.5% isoflurane (Hanapharm. Co., Ltd., Seoul, Republic of Korea) in 100% oxygen. The anesthetized rats were then secured on an acrylic platform raised at a 45° angle to allow for the flow of blood. The abdominal area was then washed and disinfected using 70% ethanol. A lengthwise abdominal incision was made to expose the liver organ after which a laminated card was inserted under the left lateral lobe of the liver. While recording video footage, a 4.0 mm biopsy punch (Kai Medical, Seki, Japan) was used to puncture the exposed liver lobe. The punctured liver was allowed to bleed for at least 30 s after which 100 µL of the test sample was applied using a 1 mL syringe (Sungshim, Bucheon, Republic of Korea) with a 26 gauge needle. Blood flow on the laminated card was routinely wiped off to assess if the liver puncture was still bleeding. The bleeding times were recorded by reviewing the video footage, and the data were tabulated and statistically analyzed.

### 4.3. In Vivo Wound Healing

The impact of applying a gel-type antiseptic on wound healing was examined using an animal model with a full-thickness skin defect. Twenty Sprague Dawley rats (Orientbio Co., Republic of Korea) of similar age and weight from the hemostatic testing were randomly distributed to 5 treatment groups (*n* = 4). The animals were housed in pairs in the same animal room. Food and water were provided ad libitum during the length of the entire experiment period. Anesthetic induction was performed on rats using the aforementioned isoflurane inhalation. Fur from the rats’ dorsal side was cleared by shaving and the application of a hair removal cream. Ethanol solution (70%) was used to clean the bare skin surface and clear any excess hair removal cream. The spine of the animal was identified, and two identical full-thickness skin defects were created on both sides of the animal using a 10.0 mm biopsy punch. The wound surfaces were treated with 1.0 mL of test samples (PI3-S/PI3-G/PI3-G+TMC/PI3-LH) and allowed to soak for 2 min. A group treated with normal saline solution served as a positive control reference. The animals were then returned to the housing cages and allowed to recover. Test samples were routinely applied every 2 days. Images of the wound surfaces were taken to measure and determine the wound size progression. Tissue samples were extracted after 8 days and fixed using 10% neutral buffered formalin (Mirax, Suwon, Republic of Korea). Formalin-fixed skin tissue samples were then processed for paraffin embedding using a Tissue-Tek VIP^®^5 Jr. (Sakura Finetek, Torrance, CA, USA) automated processing machine. Paraffin-embedded samples were then cut into 5.0 µm sections using a Leica RM2135 microtome (Leica, Wetzlar, Germany) and stained with hematoxylin and eosin. Images of the stained tissue sections were taken using an Olympus BX53 light microscope with cellSens imaging software (version 1.1, Olympus Life Science, Tokyo, Japan). These images were imported into FIJI software (version 2.9.0) for histomorphometric analyses. Images from each test group were used to compare the number of blood vessels, fibrous tissue thickness, and epidermal tissue thickness. Measurements were tabulated and statistically compared with among the different test groups.

### 4.4. Statistical Analyses

Statistical analyses were performed using GraphPad Prism 8.0 (GraphPad Software Inc., La Jolla, CA, USA). The data were subjected to one-way analysis of variance (ANOVA), followed by Tukey’s multiple comparison tests to identify significant variations among treatment groups, with the significance level set at *p* < 0.05.

## 5. Conclusions

In conclusion, this comprehensive study provides critical validation for strategically incorporating TMC into commercially available formulations of PVP-I for wound management applications. The in-depth exploration of its antimicrobial properties, conducted through rigorous in vitro testing revealed that the inclusion of TMC did not compromise the established potency of PVP-I against various bacterial species and a fungus. This reassuring finding not only underscores the compatibility of TMC with PVP-I but also emphasizes its potential as a valuable additive for antimicrobial synergy. Beyond the realm of antimicrobial efficacy, the study delved into the hemostatic capabilities of the TMC-enhanced PVP-I gel, showcasing a noteworthy improvement that positions it to be on par with the effectiveness of PVP-I liposomal hydrogel. The convergence of antimicrobial and hemostatic attributes in a single formulation holds promising implications for wound management, especially in scenarios where both infection control and bleeding management are pivotal concerns. Although the application of the TMC-containing PVP-I gel did not result in significant enhancement in wound size reduction the regenerated dermal tissue exhibited a relatively advanced stage of development, surpassing the outcomes associated with conventional products. This nuanced observation suggests that the incorporation of TMC contributes to the intricate processes of tissue regeneration, potentially influencing cellular behavior and signaling factors crucial for optimal wound healing.

## Figures and Tables

**Figure 1 ijms-25-02106-f001:**
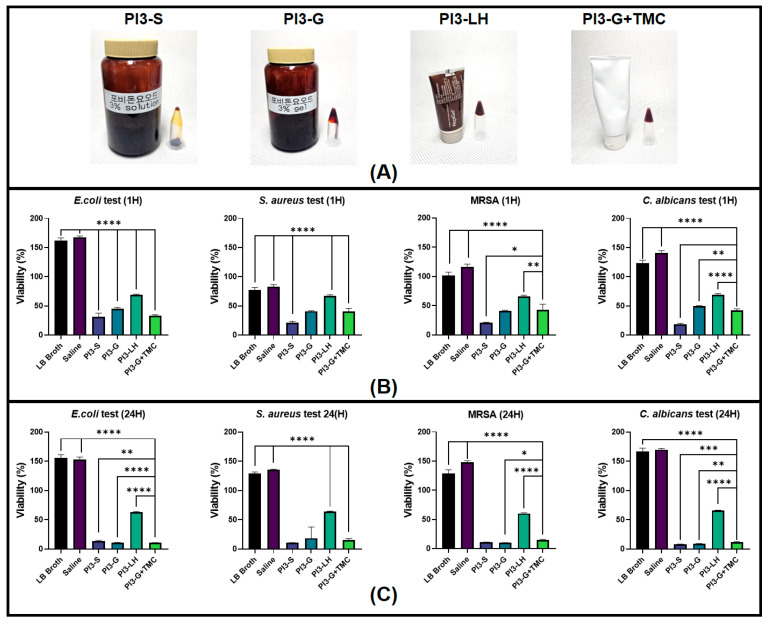
Images of the different Poly(vinylpyrrolidone)-iodine (PVP-I) formulations (**A**). Viability of bacterial and fungal cultures with the test materials after 1 h (**B**) and 24 h (**C**) (*n* = 6; * *p* < 0.05; ** *p* < 0.01; *** *p* < 0.001; **** *p* < 0.0001).

**Figure 2 ijms-25-02106-f002:**
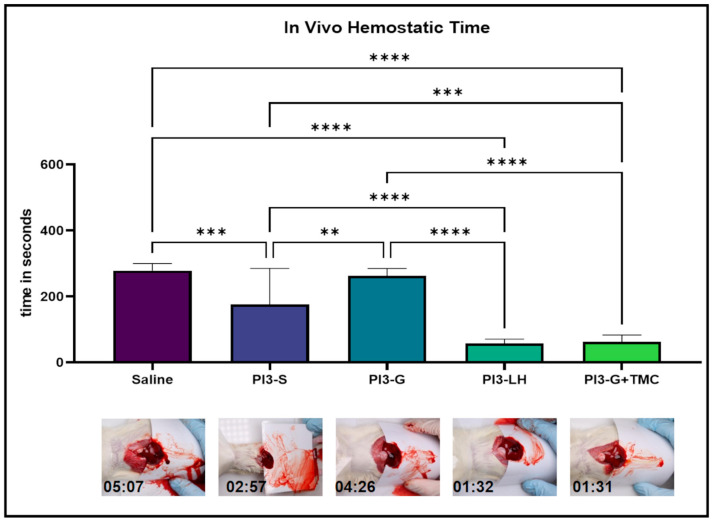
In vivo bleeding experiments testing the hemostatic activity of the different samples. Averaged timing scores of each group indicate that PI3-S and PI3 g were not suitable for stopping the bleeding, while both PI3-LH and PI3-G+TMC were effective in minimizing the bleeding time of a liver puncture (*n* = 5; ** *p* < 0.01; *** *p* < 0.001; **** *p* < 0.0001).

**Figure 3 ijms-25-02106-f003:**
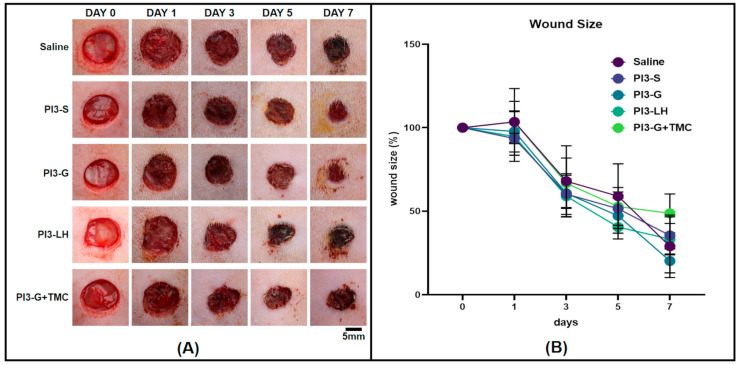
Macro images of the rat full-thickness skin wounds taken over the period of 1 week (**A**) and averaged percentage measurement of the wound areas per treatment group (**B**) (*n* = 4).

**Figure 4 ijms-25-02106-f004:**
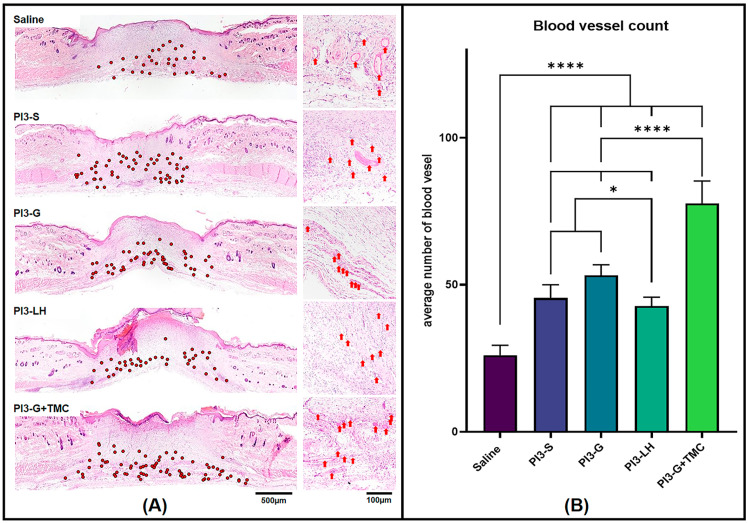
Hematoxylin- and eosin-stained tissue section (**A**) revealed varied presence of blood vessels within the wound area (Low magnification = red dots; high magnification = red arrows). Histomorphometric count of blood vessels within the wound area (**B**) indicate the significantly higher presence of blood vessels in PI3-G+TMC-treated wound compared to the other groups (*n* = 4; * *p* < 0.05; **** *p* < 0.0001).

**Figure 5 ijms-25-02106-f005:**
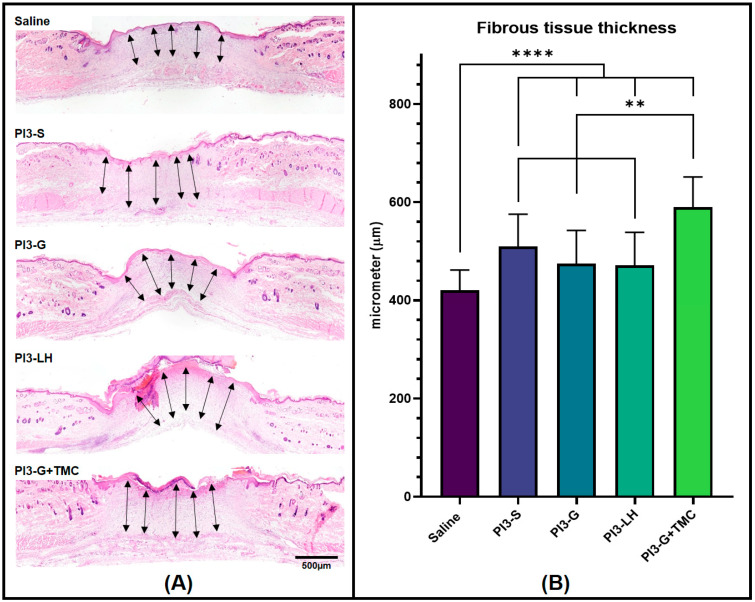
Histological images showing the fibrous tissue formed within the wound bed of the skin defects (**A**) and histomorphometric measurement of the fibrous tissue thickness (**B**). Arrows indicate measured cross-sectional lengths (*n* = 4; ** *p* < 0.01; **** *p* < 0.0001).

**Figure 6 ijms-25-02106-f006:**
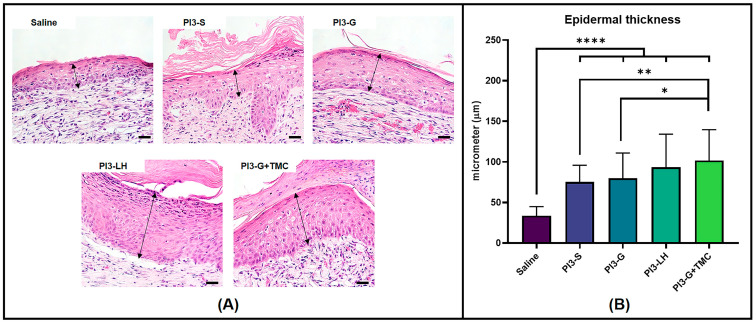
High magnification images showing the epidermal thickness formed atop the fibrous tissue within the wound bed (**A**) (Arrows indicate measured cross-sectional thickness; scale bar at 20 µm). Histomorphometric measurements (**B**) of epidermal tissue thickness measured from regenerated wounds (*n* = 4; * *p* < 0.05; ** *p* < 0.01; **** *p* < 0.0001).

## Data Availability

Data are contained within the article or Appendix A.

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
