# Peer review of "Antiseptic, Hemostatic, and Wound Activity of Poly(vinylpyrrolidone)-Iodine Gel with Trimethyl Chitosan"

_ijms, 2024, doi:10.3390/ijms25042106_

Round 1

Reviewer 1 Report

Comments and Suggestions for Authors

1. Would mixing in some antimicrobial drug, e.g., chlorhexidine acetate, provide a greater advantage than mixing in chitosan? Or, does chitosan significantly enhance the physicochemical and biological functions of PVP-I, which cannot be replaced by drugs?

2. Does the introduction of chitosan add another significant enhancement to the antimicrobial properties? This is something the authors need to show.

3. Background descriptions for hydrogel dressings can be strengthened by citing 10.1016/j.cej.2022.135691; 10.1016/j.carbpol.2021.118046.

4. There are some formatting errors in the article. For example, space is required between the number and the character. Please check the dash and hyphen line used in this article. Kindly check for correctness.

5. The Abstract section should be strengthened: the important results and main conclusions drawn in this paper should be highlighted and presented in more precise language.

Author Response

Thank you for giving us the chance to address your concerns regarding our submission. We have modified the manuscript based on your comments and recommendations. Changes in the revised manuscript have been highlighted in yellow and are quoted in our respective responses to your specific questions.

  1.  
  2. Would mixing in some antimicrobial drug, e.g., chlorhexidine acetate, provide a greater advantage than mixing in chitosan? Or, does chitosan significantly enhance the physicochemical and biological functions of PVP-I, which cannot be replaced by drugs?

Response: We acknowledge your concern and we have included additional statements in the discussion to emphasize the advantage of incorporating TMC with PVP-I gel.

  • “Studies of late have demonstrated the versatility of TMC as either a drug carrier [78, 79] or a scaffold component [80, 81]. Only a few studies have been focused on investigating the wound- healing effect of TMC. Notably, it has been used as a carrier of DNA plasmid encoding for vascularization growth factors to promote angiogenesis in full-thickness skin defects [82-84]. Owing to its innate biocompatibility, chitosan and chitosan derivatives such as TMC have been proven to advance the growth of granulation tissue crucial for wound healing and vascularization [85, 86].”
  1. Does the introduction of chitosan add another significant enhancement to the antimicrobial properties? This is something the authors need to show.

Response: We appreciate your comment and we have included additional statements about this point of discussion. As stated in the discussion section, the addition of the TMC did not significantly improve or reduce the efficacy antimicrobial activity of the resulting antiseptic agent. Thus, the use of TMC can be considered as an alternative to chlorhexidine or alcohol in conjunction with the application of PVP-I since it offers a different mode of antimicrobial activity with considerably less toxicity when compared to chlorhexidine or alcohol.

  • “Although TMC also has an antibacterial effect [37, 42-44], its addition to the gel type PVP-I did not further improve its antimicrobial properties. This establishes that the incorporation of TMC in PVP-I gel did not affect its bactericidal and fungicidal capability even when tested under high microbial load. This is quite similar if not less notable compared to other types of antiseptics that have also been used with PVP-I -chlorhexidine and alcohol. These combinations are meant to provide dual modes of antimicrobial activity for antiseptic applications. Previous studies have indicated that TMC confers its antimicrobial activity through its polycationic character, allowing it to highly interact and disrupt microbial cytoplasmic membranes and walls [53-55]. This type of mode of action is quite similar to that of chlorhexidine, making both effective against gram - positive bacteria, gram -negative - bacteria, fungi, and viruses [55-57]. On the other hand, the antimicrobial activity of alcohol is based on its denaturation of proteins making it more effective in when diluted with water [58, 59]. However, unlike chlorhexidine which can be deactivated upon interaction with anionic compounds and surfactants [60, 61], TMC constantly holds a positive charge imparting relatively prolonged antiseptic activity [53, 54, 62]. Similarly, chitosan and chitosan derivatives are less toxic [37, 63] to mammalian cells compared to both alcohol of chlorhexidine, allowing for its utilization at higher concentrations which can further enhance its antimicrobial activity in a dose - dependent manner.”
  1. Background descriptions for hydrogel dressings can be strengthened by citing 10.1016/j.cej.2022.135691; 10.1016/j.carbpol.2021.118046.

Response: We have added this reference in the introduction section. 

  1. There are some formatting errors in the article. For example, space is required between the number and the character. Please check the dash and hyphen line used in this article. Kindly check for correctness.

Response: We have scanned the whole manuscript and adjusted spacings and punctuations based on the journal’s format. (Please see highlighted sections of the revised manuscript).

  1. The Abstract section should be strengthened: the important results and main conclusions drawn in this paper should be highlighted and presented in more precise language.

Response: Thank you for this recommendation. We have modified the abstract section to highlight the conclusions that were drawn from the results of the study.

  • “Wound management practices have advanced significantly, but the quest for improved antiseptics continues. Seeking solutions that not only prevent infections but also address broader aspects of wound care, we examined the impact of integrating trimethyl chitosan (TMC) into a widely used poly (vinylpyrrolidone)-iodine gel (PVP-I gel). Our study evaluated the antimicrobial efficacy of the PVP gel with TMC against Escherichia coli, Staphylococcus aureus, multi-drug-resistant S. aureus MRSA, and Candida albicans. Additionally, we compared the hemostatic effects using a liver puncture bleeding model and assessed wound healing through histological sections from full-thickness dermal wounds in rats. Results indicate that incorporating TMC into the commercially available PVP-I gel did not compromise its antimicrobial activity. Regular application of the PVP-I gel with TMC significantly enhanced hemostatic activity, increased blood vessel count in the wound bed, and facilitated the development of thicker fibrous tissue and a regenerated epidermal layer. These findings suggest that TMC contributes not only to antimicrobial activity but also to the intricate processes of tissue regeneration. In conclusion, incorporating TMC proves beneficial, making it a valuable additive to commercially available antiseptic agents.”

Reviewer 2 Report

Comments and Suggestions for Authors

Interesting paper on the comparison of PVP based gel on wound healing. The study include antibacterial in vitro study, an haemostasis study in livers and in vivo skin studies looking at the macroscopic regeneration of the skin and, microscopically, at the quality of the newly formed tissue.

Tbe abstract present grammatical mistakes.

There are large portions of the introduction with no references. Make sure to use more citations to support all the statements.

All the materials mustreporter the manufacturer and the city and country of production. It must be possible for other researchers to reproduce your work.

For the haemostasis experiment, how did you make sure to apply a consistent amount of treatment? Have the formulations had the same viscosity, rheological properties?

Were thee mice sacrificed after the procedure?

Line 373 doesn't make sense, check the grammar.

How were animals housed? Were kept hydrated and fed throughout the experiment? Were they kept isolated?

The tissue was fixed, and was it embedded? If yes, how? What was the thickness of the sections? 

I think you must indicate the different elements you are counting on the skin sections. Use arrows to highlight the epidermis, larger images to highlight the blood vessels, etc. Remember that not all the readers are expert.

For the in vitro antibacterial activity, have you tested the effects of the treatments only on the MTT activity? Some of these substance might interfere with the assay. Was the culture kept in static or dynamic conditions (with shaking?). This might have had an influence on the results and it would be worth to explore.

Comments on the Quality of English Language

Overall the English is correctly used, but there are some grammatical and syntax mistakes to address.

Author Response

Response to Reviewer Comments

Reviewer 2

Interesting paper on the comparison of PVP based gel on wound healing. The study include antibacterial in vitro study, an haemostasis study in livers and in vivo skin studies looking at the macroscopic regeneration of the skin and, microscopically, at the quality of the newly formed tissue.

Response: Thank you for giving us the chance to address your concerns regarding our submission. We have modified the manuscript based on your comments and recommendations. Changes in the revised manuscript have been highlighted in yellow and are quoted in our respective responses to your specific questions.

  1. The abstract present grammatical mistakes.

Response: We have revised the abstract section to improve its readability and highlight the conclusions that were drawn from the results of the study

  • “Wound management practices have advanced significantly, but the quest for improved antiseptics continues. Seeking solutions that not only prevent infections but also address broader aspects of wound care, we examined the impact of integrating trimethyl chitosan (TMC) into a widely used poly (vinylpyrrolidone)-iodine gel (PVP-I gel). Our study evaluated the antimicrobial efficacy of the PVP gel with TMC against Escherichia coli, Staphylococcus aureus, multi-drug-resistant S. aureus MRSA, and Candida albicans. Additionally, we compared the hemostatic effects using a liver puncture bleeding model and assessed wound healing through histological sections from full-thickness dermal wounds in rats. Results indicate that incorporating TMC into the commercially available PVP-I gel did not compromise its antimicrobial activity. Regular application of the PVP-I gel with TMC significantly enhanced hemostatic activity, increased blood vessel count in the wound bed, and facilitated the development of thicker fibrous tissue and a regenerated epidermal layer. These findings suggest that TMC contributes not only to antimicrobial activity but also to the intricate processes of tissue regeneration. In conclusion, incorporating TMC proves beneficial, making it a valuable additive to commercially available antiseptic agents.”
  1. There are large portions of the introduction with no references. Make sure to use more citations to support all the statements.

Response: We have added citations to several statements in the introduction section. (Please see highlighted sections of the revised manuscript)

  1. All the materials mustreporter the manufacturer and the city and country of production. It must be possible for other researchers to reproduce your work.

Response: We have indicated additional information in the methods sections. The company and country of origin were added to the materials used for each experiment. (Please see highlighted sections of the revised manuscript)

  1. For the haemostasis experiment, how did you make sure to apply a consistent amount of treatment? Have the formulations had the same viscosity, rheological properties?

Response: We have provided additional details in the methods section for a better description of the experiment procedure. As stated, we applied 100 µL of each test sample using a 1 mL syringe with a 26-gauge needle. The different formulations did not have similar viscosity as shown in Figure 1 A.

  • “ While video recording, a 4.0 mm biopsy punch (Kai Medical, Japan) was used to puncture the exposed liver lobe. The punctured liver was allowed to bleed for at least 30 seconds after which 100 µL of the test sample was applied using a 1 mL syringe (Sungshim, Korea) with a 26 gauge needle”
  1. Were thee mice sacrificed after the procedure?

Response: We have indicated the number of animals for both hemostasis and wound healing experiments in the methods section.

  • “For the hemostasis test, 25 Sprague Dawley male rats weighing approximately 200 – 250 g (6 weeks old, Orientbio Co., South Korea) rats were randomly distributed among 5 treatment groups (n = 5), one of which was treated with saline as a positive control reference”
  • “The effect of the application of gel-type antiseptic on wound healing was tested using an animal model with a full-thickness skin defect. Twenty Sprague Dawley (Orientbio Co., South Korea) of similar age and weight from the hemostatic testing were randomly distributed to 5 treatment groups (n = 4).”
  1. Line 373 doesn't make sense, check the grammar.

Response: This statement has been modified for clarity and correctness.

  • “Tissue samples were extracted after 8 days and fixed using 10% neutral buffered formalin (Mirax, Korea). Formalin- fixed skin tissue samples were then processed for paraffin embedding using a Tissue-Tek VIP®5 Jr. (Sakura Finetek, USA) automated processing machine. Paraffin- embedded samples were then cut into 5.0 µm sections using Leica RM2135 microtome (Leica, Germany) and stained with hematoxylin and eosin. Images of the stained tissue sections were taken using an Olympus BX53 light microscope with cellSens imaging software (Build 18987, Olympus Life Science, Tokyo Japan). Images were imported to FIJI software (ImageJ) for histomorphometric analyses.”
  1. How were animals housed? Were kept hydrated and fed throughout the experiment? Were they kept isolated?

Response: The animals were housed in groups of three rats for the hemostatic test experiment and in pairs for the hound healing experiment. We have provided additional details regarding animal housing and maintenance in the methods section.

  • “For the hemostasis test, 25 Sprague Dawley male rats weighing approximately 200 – 250 g (6 weeks old, Orientbio Co., South Korea) rats were randomly distributed among 5 treatment groups (n = 5), one of which was treated with saline as a positive control reference. Three rats were housed per cage and placed in climate-controlled animal room with a 12 - hour light-dark cycle. Food and water were provided constantly. The hemostasis experiment was conducted after 1 week of acclimatization.”
  • “Twenty Sprague Dawley (Orientbio Co., South Korea) of similar age and weight from the hemostatic testing were randomly distributed to 5 treatment groups (n = 4). The animals were housed in pairs in the same animal room. Food and water were provided ad libitum during the length of the entire experiment period.”
  1. The tissue was fixed, and was it embedded? If yes, how? What was the thickness of the sections? 

Response:  We have modified the methods section to better state the technique used for the preparation of tissue sections.

  • “Tissue samples were extracted after 8 days and fixed using 10% neutral buffered formalin (Mirax, Korea). Formalin- fixed skin tissue samples were then processed for paraffin embedding using a Tissue-Tek VIP®5 Jr. (Sakura Finetek, USA) automated processing machine. Paraffin- embedded samples were then cut into 5.0 µm sections using Leica RM2135 microtome (Leica, Germany) and stained with hematoxylin and eosin. Images of the stained tissue sections were taken using an Olympus BX53 light microscope with cellSens imaging software (Build 18987, Olympus Life Science, Tokyo Japan). Images were imported to FIJI software (ImageJ) for histomorphometric analyses.”
  1. I think you must indicate the different elements you are counting on the skin sections. Use arrows to highlight the epidermis, larger images to highlight the blood vessels, etc. Remember that not all the readers are expert.

Response: We have modified the image in Figure 4 to better visualize the distribution of the blood vessels identified in the representative tissue sections. We have also added a higher magnification image showing the identified blood vessels per treatment. Please see Figure 4 (A).

  1. For the in vitro antibacterial activity, have you tested the effects of the treatments only on the MTT activity? Some of these substance might interfere with the assay. Was the culture kept in static or dynamic conditions (with shaking?). This might have had an influence on the results and it would be worth to explore.

Response: We appreciate your curiosity regarding this test.  We acknowledge that the PVP-I could interfere with the absorbance reading from the MTT test, thus during the experiment, we also prepared a blank well containing only the samples without the microbial inoculates. Readings from the respective blank wells of each sample were then subtracted from MTT absorbance values in the test wells seeded with bacteria/fungi. This allowed us to remove any readings from the respective samples from the true absorbance readings coming from the MTT incubation. We have provided additional details in the methods section for clarification. All culture conditions were done under static conditions since this allowed the simulation of a surface covered with bacteria/ fungi similar to infected wounds. Our previous optimization tests for this experiment indicated that shaking did not result in significantly different readings compared to the static culture method.

  • “Viability tests were conducted based on previously established protocol [40, 87, 88] with slight modification. Briefly, inoculations were done in a 96-well culture plate by mixing 100 µL of LB Broth (Sigma-Aldrich, USA), 40 µL of bacterial/fungal solution, and 40 µL of test sample per well (n = 6) which was then placed in a 37° C incubator. Blank wells containing 140 µL of LB broth and 40 µL of test sample were also prepared as negative controls. The amount of bacteria/fungus that survived after 1 hour and 24 hours of incubation was determined by adding 20 µL of thiazolyl blue tetrazolium bromide (MTT) (Sig-ma-Aldrich, USA) solution at a concentration of 5.0 mg/mL. Absorbance values of the plated bacterial /fungus cultures were measured after shaking at 600 nm using a TECAN InfiniteM200 Pro (Tecan Group Ltd., Zürich, Switzerland) plate reader. All absorbance values were normalized by subtracting the readings from the respective blank wells of each test sample. Absorbance values of plated bacterial cultures immediately incubated with MTT solution were used as the reference baseline viability. Percent viability was calculated by dividing the normalized absorption data of each   sample by the averaged baseline absorbance values.”

Round 2

Reviewer 2 Report

Comments and Suggestions for Authors

Figure 4 is missing.

Figure 3 reports arrows and points, but nothing is said about what they are in the legend.

Comments on the Quality of English Language

Minor English editing needed

Author Response

Reviewer 2

Thank you for providing your critical input. We have edited the manuscript based on your recommendations and queries. We have also modified the manuscript to remove grammatical errors and improve its overall readability.

  1. Figure 4 is missing.

Response: We apologize for this mistake. We have provided the proper images for all figures. Please see the revised manuscript.

  1. Figure 3 reports arrows and points, but nothing is said about what they are in the legend.

Response: Again, apologize for this mistake. The image indicated in Figure 3 was intended for Figure 4. We have provided the proper images for all figures in the manuscript. Please see the revised manuscript.